# Silicon fertigation and salicylic acid foliar spraying mitigate ammonium deficiency and toxicity in *Eucalyptus* spp. clonal seedlings

**Jonas Pereira de Souza Junior**[1]*, **Renato de Mello Prado**[1], **Thaís chagas Barros de Morais**[1], **Joaquim José Frazão**[1], **Marcilene Machado dos Santos Sarah**[1], **Kevein Ruas de Oliveira**[2], **Rinaldo César de Paula**[3]

**1** Laboratory of Plant Nutrition, Department of Agricultural Production Science, São Paulo State University (UNESP), Jaboticabal, São Paulo, Brazil, **2** Laboratory of Plant Physiology, Department of Biology Applied to Agriculture, São Paulo State University (UNESP), Jaboticabal, São Paulo, Brazil, **3** Laboratory of Forestry, Department of Plant Production, São Paulo State University (UNESP), Jaboticabal, São Paulo, Brazil

* jonas.psj@hotmail.com

**Data Availability Statement:** All relevant data are within the manuscript.

## Abstract

Nitrogen deficiency and toxicity, primarily in its ammonium form ($NH_4^+$), can suppress plant growth and development. The use of silicon (Si) or salicylic acid (SA) may be an alternative to minimize the harmful effects of nutrient imbalances caused by $NH_4^+$, thereby improving the photosynthetic efficiency of plants. The aim of the present study was to assess the action of fertigation-applied Si and SA foliar spraying in mitigating $NH_4^+$ toxicity and deficiency in eucalyptus clonal seedlings. Two experiments were performed with eucalyptus clonal seedlings (*Eucalyptus urophylla* x *Eucalyptus grandis*), in a greenhouse. Both were carried out using a 4x2 factorial design and four concentrations of $NH_4^+$ (5, 15, 30 and 60 mmol $L^{-1}$), in the absence and presence of Si (2 mmol $L^{-1}$), in experiment I; or with and without SA foliar application ($10^{-2}$ mmol $L^{-1}$), in experiment II, with six repetitions. Nitrogen content rose as a result of increasing $N-NH_4^+$ concentration in the nutrient solution, and Si supplied via the nutrient solution was efficient in increasing the Si content in eucalyptus seedlings. The rise in $N-NH_4^+$ concentration favored the maintenance of the photosynthetic apparatus, but high $N-NH_4^+$ concentration increased energy loss through fluorescence and decreased the efficiency of photosystem II. The addition of Si to the nutrient solution proved to be beneficial to the photosynthetic apparatus by decreasing $F_0$ at 15 and 30 mmol $L^{-1}$ of $NH_4^+$; and $F_m$ at all $NH_4^+$ concentrations studied. In addition, the beneficial element also increases $F_v/F_m$ at all $NH_4^+$ concentrations studied. SA foliar application was also efficient in reducing photosynthetic energy losses by decreasing $F_0$ and $F_m$ at all $NH_4^+$ concentrations studied. However, SA only increased the $F_v/F_m$ at the high concentrations studied (30 and 60 mmol $L^{-1}$ of $NH_4^+$). Nitrogen disorder by deficiency or $N-NH_4^+$ toxicity reduced shoot dry mass production. The addition of Si to the nutrient solution and SA foliar application increased shoot dry mass production at all $N-NH_4^+$ concentrations studied, and benefitted the photosynthetic apparatus by decreasing fluorescence and improving the quantum efficiency of photosystem II as well as dry mass production.

**Funding:** we do not receive specific funds for this work.

**Competing interests:** The authors have declared that no competing interests exist.

## 1. Introduction

In Brazil, eucalyptus, one of the most widely planted forest species, is being used to replace pastures or renew of old plantations, whose acreage continues to expand [1]. This increases the demand for more tree planting [2], and the need for greater seedling production. Adequate plant nutrition is essential to produce quality seedlings, particularly nitrogen (N), an important element for the development and production of biomass, since N deficiency decreases photosynthesis and biomass accumulation in eucalyptus seedlings [3].

Plant N uptake can occur in the form of nitrate ($NO_3^-$) and ammonium ($NH_4^+$); however $NH_4^+$ uptake is preferred for plants grown in nutrient solution [4]. Adequate $NH_4^+$ uptake favors plants because the nutrient is already in a reduced form that allows its incorporation into carbon skeletons without the need to reduce the N that occurs with the $NO_3^-$ uptake [5]. Despite the beneficial effect of supplying N as $NH_4^+$ to plants, high concentrations of $NH_4^+$ can suppress plant growth and development [6, 7].

A possible alternative to mitigate the effect of N disorder in plants is the use of silicon (Si). Although not considered a plant nutrient, this element has beneficial effects in terms of mitigating stress caused by abiotic factors such as nutritional imbalances, improving photosynthetic efficiency and biomass production in a number of species cultivated under N deficiency [8, 9] or N-$NH_4^+$ toxicity [6, 10].

Another alternative to mitigate N disorder is the application of salicylic acid (SA). Belonging to the group of phenolic compounds, SA can be found in the leaves and reproductive structures of plants [11]. Considered a plant hormone, it participates in regulating the physiological processes, favoring photosynthesis and growth [12–14], processes compromised in cases of N deficiency and N-$NH_4^+$ toxicity.

In this respect, it is hypothesized that Si and SA can mitigate N-deficiency and N-toxicity, in its $NH_4^+$ form, in eucalyptus seedlings, by increasing photosynthetic efficiency. If confirmed, this hypothesis will contribute to broadening the possibilities of Si and SA alternative use as N-$NH_4^+$ disorder mitigators, which may optimize eucalyptus seedling production.

Thus, the aim of the present study was to assess the action of fertigation-supplied Si and SA foliar spraying in photosynthetic efficiency and biomass production, and as mitigators of N deficiency and N toxicity, in its $NH_4^+$ form, in eucalyptus clonal seedlings.

## 2. Materials and methods

### 2.1 Plant material and growth conditions

Two experiments were carried out in a greenhouse with eucalyptus clonal seedlings (*Eucalyptus urophylla* x *Eucalyptus grandis*, clone H13). In both experiments, the roots of recently rooted eucalyptus seedlings were washed, and one seedling was placed in 50 $cm^3$ tubes filled with medium-textured vermiculite and kept a under mist spray irrigation system for two weeks, in order to ensure good establishment.

Next, the seedlings of the two experiments received a nutrient solution without N [15], with pH between 5.5 and 6.5, and Fe-EDDHMA as iron source. The nutrient solution was provided to plants at 25% of the concentration indicated by the authors for 7 days. After this period, the concentration was increased to 50%, which was maintained throughout the experiment period. The nutrient solution was supplied twice a day, with 10 mL applied in the early morning and 10 mL in the late afternoon. The substrate was washed daily to eliminate excess salt. During substrate drainage, 20 mL of deionized water was added to each pot, inducing drainage of 10 mL of nutrient solution, which was discarded. After 2h, a new nutrient solution was supplied

to the plants and during the rest of the crop cycle the nutrients needed for plant development were provided.

## 2.2 Experiment I

Experiment I was carried out in a completely randomized factorial design (4 x 2): with four $NH_4^+$ concentrations (5, 15, 30 and 60 mmol $L^{-1}$) in the absence and presence (2 mmol $L^{-1}$) of Si, and six repetitions of two seedlings each. This Si concentration (2 mmol $L^{-1}$) was used because it is known that polymerization starts decreasing Si monomer species (monosilicic acid) at higher concentrations, thereby reducing Si absorption by plants [16].

Ammonium chloride was used as ammonium source. To establish ammonium concentration, 15 mmol $L^{-1}$ of N was used as reference [15], applying one-third (5 mmol $L^{-1}$ of N), double (30 mmol $L^{-1}$ of N) and quadruple (60 mmol $L^{-1}$ of N) this amount.

The silicate solution was prepared from potassium silicate (SiK), adjusting pH to between 5.5 and 6.5 using an HCl (1 mol $L^{-1}$) and NaOH (1 mol $L^{-1}$) solution, supplied to the eucalyptus seedlings by fertigation in the first 7 days after transplanting. Next, Si was added to the nutrient solution and immediately supplied to the seedlings via fertigation, with pH maintained between 5.5 and 6.5. Since the Si source (SiK) contained potassium (K), the concentration of this element was balanced between the treatments with and without Si.

## 2.3 Experiment II

Experiment II was also carried out in a completely randomized factorial design (4x2): with the same four $NH_4^+$ concentrations (5, 15, 30 and 60 mmol $L^{-1}$), applied as ammonium chloride, in the absence and presence ($10^{-2}$ mmol $L^{-1}$) of SA, with six repetitions of three plants each. This SA concentration ($10^{-2}$ mmol $L^{-1}$) was previously tested in eucalyptus clonal seedlings. Laboratory tests were carried out with increasing concentrations of SA to determine the highest beneficial concentration of this element for plants without causing negative effects such as plant hormones imbalance [17].

A solution with SA was prepared and three foliar applications were made when the plants exhibited 4, 6 and 8 pairs of fully expanded leaves. For the SA solution, the pH of the deionized water was raised to between 11 and 12 using NaOH (1 mol $L^{-1}$), in order to solubilize the SA. At the moment of foliar application, the pH of the solution was adjusted to between 6.5 and 7.0 using HCl (1 mol $L^{-1}$). The solution was applied with a handheld sprayer at 0.5, 1.0 and 1.5 mL of solution per plant, for the first, second and third application, respectively. The plants not sprayed with the SA solution received the same amount of water.

## 2.4 Analyses

### 2.4.1 Initial and maximum fluorescence, and quantum efficiency of photosystem II.

The experiments were concluded 30 days after seedling transplanting, at which time visual signs of N toxicity, in its $NH_4^+$ forms, were identified, such as chlorosis, necrosis, brown stem and wilted leaves displaying signs of senescence [18]. On this occasion, in both experiments, the initial ($F_0$) and maximum ($F_m$) fluorescence and quantum efficiency of photosystem II ($F_v/F_m$) on the second pair of fully expanded leaves were measured. These variables were obtained by measuring chlorophyll fluorescence in twelve leaves (6 repetitions and two plants per repetition) in experiment I; and eighteen leaves (6 repetitions and three plants per repetition) in experiment II, using a fluorometer (Opti-Science–OS30P). These measurements were taken at the end of the experiment, between 7:30 and 9:30 am, and on three new fully formed leaves (middle part of the stalk) per plant. Leaves were left in the dark for 30 min for adaptation purposes and then excited by a pulse of red light for 1 second, in order to determine the initial,

maximum and the variable fluorescence. The quantum efficiency of photosystem II was calculated by the ratio between maximum and variable fluorescence.

**2.4.2 Dry mass production.** After fluorescence analysis, the seedlings were collected and separated into roots and shoots. Leaves were also collected for N nutritional diagnosis (2nd and 3rd pair). The plant samples were washed with water, neutral detergent solution (0.1% v/v), HCl solution (0.3% v/v), and finally deionized water. They were then dried in a forced air circulation oven at $65 \pm 5$°C, until constant weight. Next, root and shoot dry mass were determined.

**2.4.3 Silicon and nitrogen concentration.** Plant dry mass was ground in a Willey mill, and N concentration was determined by acid digestion for both experiments, followed by distillation with NaOH and titration with $H_2SO_4$, according to the methodology described by Bataglia et al. [19]. For experiment I, Si concentration was determined by alkaline digestion with $H_2O_2$ and NaOH, followed by colorimetric (spectrophotometric) reading, as described by Kondörfer et al. [20]. Si and N accumulation were then obtained by multiplying the concentration by the corresponding dry mass.

**2.4.4 Statistical analyses.** The data were submitted to analysis of variance, followed by comparison of means (Tukey) at 5% probability. To analyze N concentrations, polynomial regression was conducted, selecting the model ($P < 0.05$) with the highest coefficient of determination ($R^2$). Statistical analysis was carried out in Sisvar® software [21].

## 3. Results

### 3.1 Analysis of variance

The rise in nitrogen concentration increased N content, Si content, $F_0$, $F_m$, $F_v/F_m$, shoot dry mass and root dry mass in *Eucalyptus* seedlings (Table 1). The presence of Si increased Si content and the presence of Si or SA decreased $F_0$, $F_m$ and $F_v/F_m$, and increased shoot and root dry mass production in *Eucalyptus* seedlings.

### 3.2 Nitrogen and silicon accumulation in eucalyptus seedlings

Nitrogen content in eucalyptus seedlings increased as a result of raising $N-NH_4^+$ concentration in the nutrient solution, peaking in experiment I at 34.4 and 42.7 g per plant at concentrations

**Table 1. Resume of analysis of variance for the mean square and coefficient of variation (CV) of nitrogen (N) content, silicon (Si) content, initial fluorescence ($F_0$), maximum fluorescence ($F_m$), quantum efficiency of photosystem II ($F_v/Fm$), shoot dry mass (SDM) and root dry mass (RDM) of *Eucalyptus* seedlings under different N concentrations in the presence of silicon (experiment I) or salicylic acid (SA–experiment II).**

| Treatments | N | Si | $F_0$ | $F_m$ | $F_v/F_m$ | SDM | RDM |
|---|---|---|---|---|---|---|---|
| **Mean Square** | | | | | | | |
| **Experiment I** | | | | | | | |
| N | 1495** | 0.68** | 1613** | 41798** | 0.02** | 1.20** | 0.55** |
| Si | 171** | 96.2** | 936** | 24396** | 0.40** | 5.38** | 6.22** |
| NxSi | 41* | 0.10ns | 167* | 1298ns | 0.01ns | 0.12ns | 0.006ns |
| **CV, %** | 7,105 | 5.3 | 5.5 | 10.7 | 4.3 | 5.6 | 10.5 |
| **Experiment II** | | | | | | | |
| N | 248.7** | - | 115** | 26003** | 0.01** | 4.24** | 0.31** |
| SA | 9.85ns | - | 118** | 16849** | 0.02** | 3.12** | 0.06ns |
| NxSA | 14.18ns | - | 3.28ns | 1204ns | 0.01** | 0.004ns | 0.009ns |
| **CV,%** | 8.3 | - | 7.1 | 5.3 | 3.14 | 6.8 | 8.3 |

**—Significant at 1% probability according to the F-test.

*—Significant at 5% probability according to the F-test.

ns–Not significant according to the F-test.

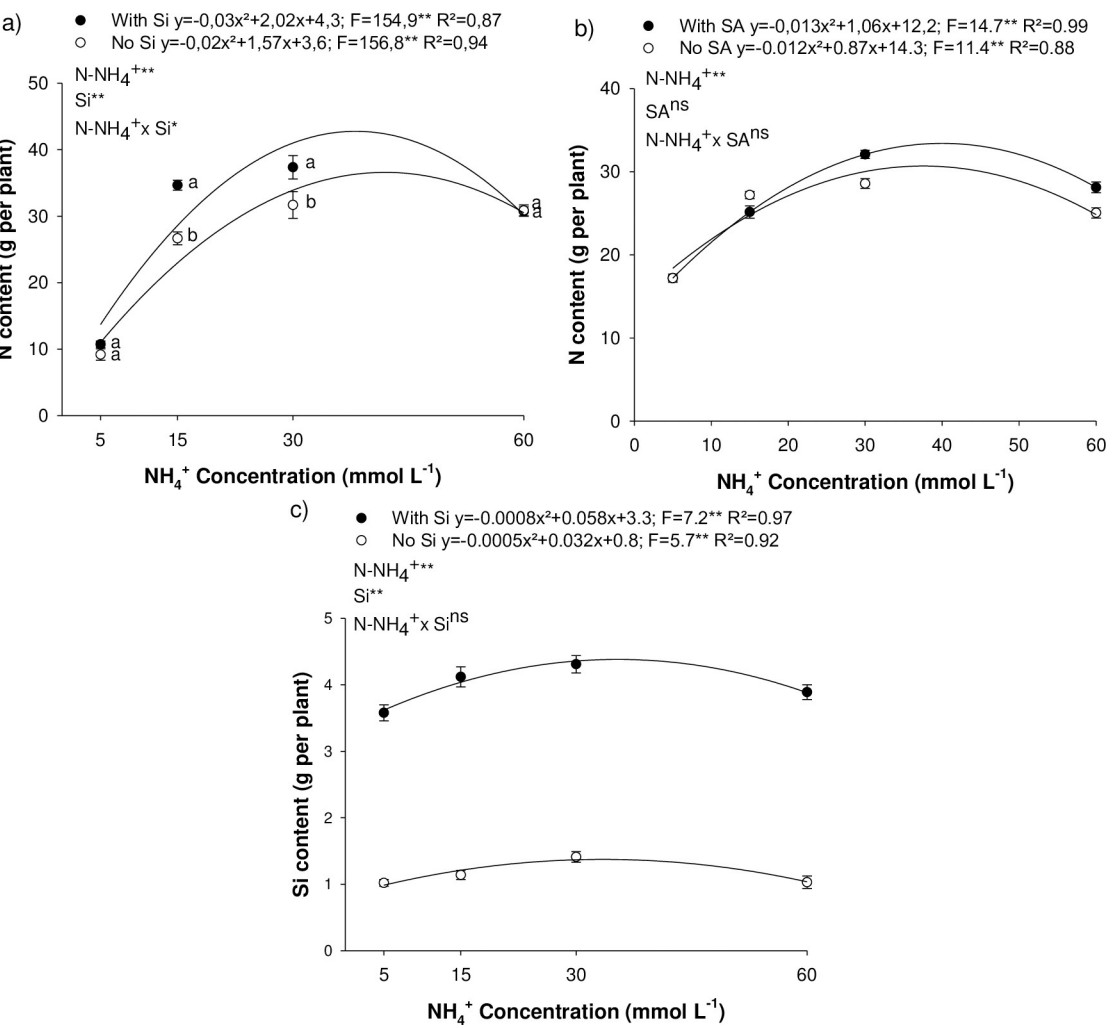

**Fig 1.** Nitrogen content in *Eucalyptus* spp. clonal seedlings as affected by increasing ammonium concentrations (N-NH$_4^+$) and the presence of silicon (Si) (a) or salicylic acid (SA) (b) presence; Silicon content in *Eucalyptus* spp. clonal seedlings as affected by increasing N-NH$_4^+$ concentrations and the presence of Si (c). ** and *: significant at 1 and 5% probability, respectively; [ns]: non-significant according to the F-test. The letters show differences between Si or AS treatments ($p < 0.05$).

of 39.2 and 33.6 g L$^{-1}$ of NH$_4^+$ in the absence and presence of Si, respectively (Fig 1A); and in experiment II, 30.1 and 33.8 g per plant at 36.2 and 40.7 g L$^{-1}$ of NH$_4^+$ in the absence and presence of Si, respectively (Fig 1B). Si supply via the nutrient solution was efficient in increasing Si content in eucalyptus seedlings, and the rise in N concentration increased Si content, peaking at 4.4 g per plant (Fig 1C). It is important to underscore that in the absence of Si, accumulation of the beneficial element was low, showing minor contamination of the nutrient solution.

## 3.3 Initial and maximum fluorescence and the quantum efficiency of photosystem II

An increase in N-NH$_4^+$ concentration favored the maintenance of the photosynthetic apparatus, indicated by the reduction in energy through fluorescence (Fig 2A–2D). Initial fluorescence decreases, as a function of N-NH$_4^+$ concentration, reaching a minimum at a

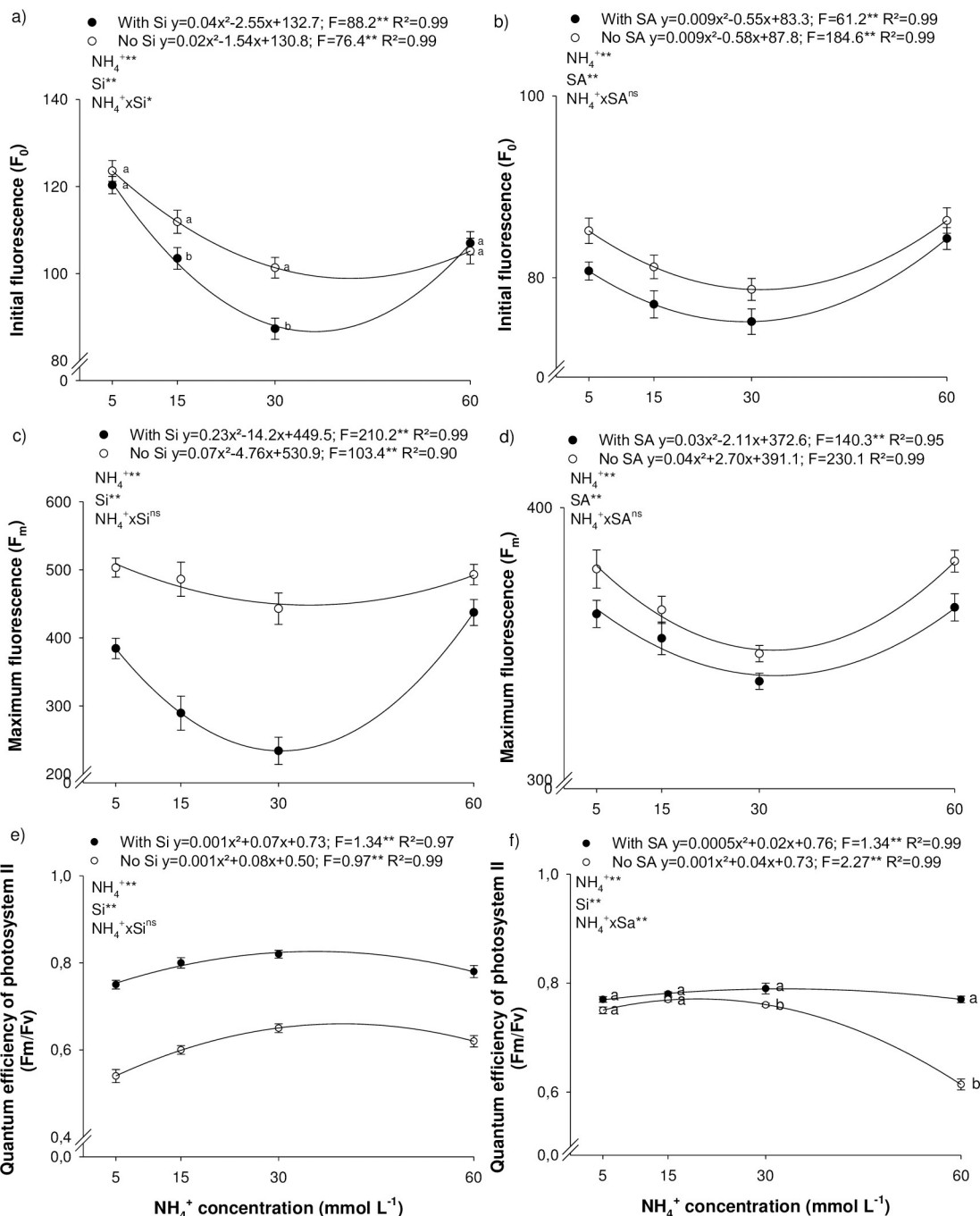

**Fig 2.** Initial fluorescence (a, b), maximum fluorescence (c, d), quantum efficiency of photosystem II (e, f) of *Eucalyptus* spp. clonal seedlings in the presence and absence of silicon (Si) or salicylic acid (SA) as a function of the increase in ammonium concentrations (N-$NH_4^+$). ** and *: significant at 1 and 5% probability, respectively; ns: non-significant according to the F-test. The letters show differences between Si or AS treatments ($p < 0.05$).

concentration of 38.5 and 31.8 g $L^{-1}$ of $NH_4^+$, in the absence and presence of Si, respectively, in experiment I (Fig 2A); and at 32.2 and 30.5 g $L^{-1}$ of $NH_4^+$, in the absence and presence of SA, respectively, in experiment II (Fig 2B). Similarly, to initial fluorescence, maximum fluorescence also decreases with a rise in $NH_4^+$ concentration. Minimum $F_m$ was observed at 34.0

and 30.8 g L$^{-1}$ of NH$_4^+$, for absence and presence of Si, respectively, in experiment I (Fig 2C); and at 33.7 and 35.2 g L$^{-1}$ of NH$_4^+$, in the absence and presence of SA, respectively, in experiment II (Fig 2D). On the other hand, the quantum efficiency of photosystem II rose with an increase in NH$_4^+$ concentrations, peaking at 0.659 and 0.825 at 40.0 and 35.0 mmol L$^{-1}$ of NH$_4^+$, in the absence and presence of Si, respectively (Fig 2E); and 0.771 and 0.785 at 20.0 and 20.0 mmol L$^{-1}$ of NH$_4^+$, for the absence and presence of SA, respectively (Fig 2F).

The addition of Si to the nutrient solution proved to be beneficial to the photosynthetic apparatus by decreasing F$_0$ at 15 and 30 mmol L$^{-1}$ of NH$_4^+$ (Fig 2A); and F$_m$ at all NH$_4^+$ concentrations studied (Fig 2C). In addition, the beneficial element also increase Fv/F$_m$ at all NH$_4^+$ concentrations studied. The SA foliar application was also efficient in decreasing photosynthetic energy losses by decreasing the F$_0$ and F$_m$ at all NH$_4^+$ concentrations studied (Fig 2B and 2D). However, SA increases F$_m$/Fv only at the high concentrations studied (30 and 60 mmol L$^{-1}$ of NH$_4^+$) (Fig 2F).

### 3.4 Eucalyptus shoot and root dry mass production

The rise in N-NH$_4^+$ concentration increased shoot dry mass production, peaking at 2.34 and 3.07 g at 34.3 and 36.1 mmol L$^{-1}$ of NH$_4^+$, in the absence and presence of Si, respectively, in experiment I (Fig 3A); and 3.06 and 3.66 g at 22.7 and 22.7 mmol L$^{-1}$ of NH$_4^+$, in the absence and presence of SA, respectively, in experiment II (Fig 3B). Nitrogen disorder caused by deficiency or N-NH$_4^+$ toxicity reduced shoot day mass production. A 15% reduction of shoot dry mass was observed at 11.6 and 53.2 mmol L$^{-1}$ of NH$_4^+$, in the absence of Si and at 13.5 and 57.0 mmol L$^{-1}$ of NH$_4^+$, in the presence of Si, in experiment I (Fig 3A). For experiment II, a shoot dry mass reduction of 15% was recorded at 42.5 and 44.8 mmol L$^{-1}$ of NH$_4^+$, for toxicity in the absence and presence of SA, respectively; and at 2.1 and 2.2 mmol L$^{-1}$ of NH$_4^+$, for deficiency in the absence and presence of SA, respectively (Fig 3B).

The addition of Si in the nutrient solution (Fig 3A) and SA foliar application (Fig 3B) increased shoot dry mass production at all N-NH$_4^+$ concentrations studied.

For roots, dry mass production rose with an increase in N-NH$_4^+$ concentration, reaching 1.25 and 1.88 g at 33.7 and 28.31 mmol L$^{-1}$ of NH$_4^+$, in the absence and presence of Si, respectively, in experiment I (Fig 3C); and 1.39 and 1.52 g at 28 and 32.5 mmol L$^{-1}$ of NH$_4^+$, in the absence and presence of SA, in experiment II (Fig 3D). A 15% decline in root dry mass production was observed at 12.5 and 45.5 mmol L$^{-1}$ of NH$_4^+$, in the absence of Si; 5.1 and 55.5 mmol L$^{-1}$ of NH$_4^+$, in the presence of Si; 7.76 and 52.4 mmol L$^{-1}$ of NH$_4^+$, in the absence of SA; and 7.57 and 52.4 mmol L$^{-1}$ of NH$_4^+$, in the presence of SA. Si supply increased root and shoot dry mass production at all N-NH$_4^+$ concentrations studied. However, no difference was observed in this variable after SA foliar application.

## 4. Discussion

NH$_4^+$ is one of the forms of N that is readily available to plants [6, 22] and easily absorbed by eucalyptus roots, increasing N accumulation, as a function of the rise in NH$_4^+$ concentrations (Fig 2A and 2B). However, high NH$_4^+$ concentrations in nutrient solution can cause toxicity, because the conversion rate into amino acids is lower than the N absorption rate, leading to an increase in NH$_4^+$ content in plant cells [6, 23]. As a result, NH$_4^+$ toxicity decreases shoot and root dry mass production (Fig 3). On the other hand, N supply at low levels decreases dry mass production (Fig 3), due to the important function of this nutrient in plant metabolism, which also occurs in eucalyptus seedlings [3].

SA foliar application does not affect N accumulation in eucalyptus plants (Fig 2B); however, Si has proved to be efficient in increasing N accumulation in several plant species [6]. Our

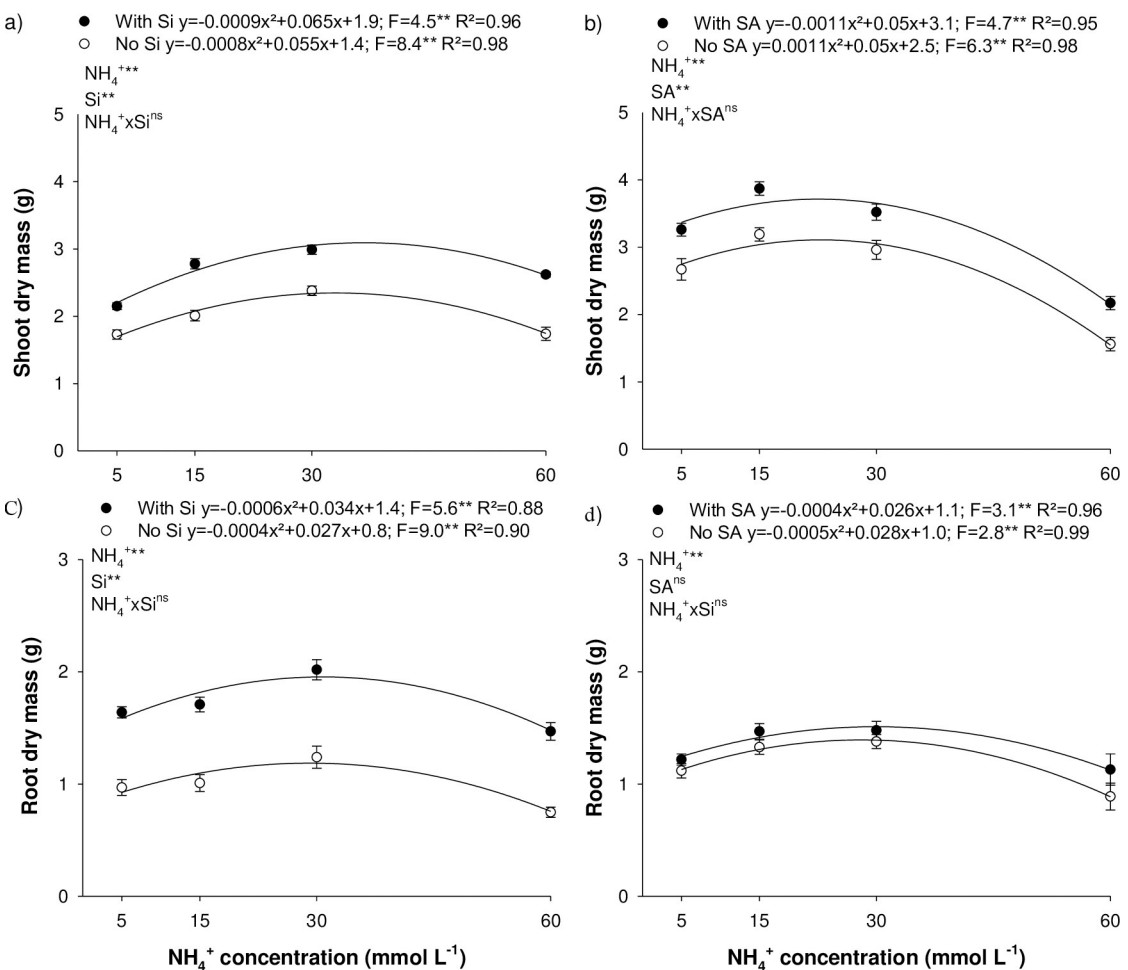

**Fig 3.** Shoot (a, b) and root dry mass (c, d) of *Eucalyptus* spp. clonal seedlings in the presence and absence of silicon (Si) or salicylic acid (SA) as a function of the increase in ammonium concentrations (N-$NH_4^+$). ** and *: significant at 1 and 5% probability, respectively; ns: non-significant according to the F-test. The letters show differences between Si or AS treatments ($p < 0.05$).

results confirmed this increased accumulation of N in eucalyptus plants under conditions of N sufficiency (15 mmol $L^{-1}$) or even at double the recommended concentration (30 mmol $L^{-1}$) (Fig 2). However, under N deficiency (5 mmol $L^{-1}$) or $NH_4^+$ toxicity (60 mmol $L^{-1}$), Si did not affect N accumulation. There results indicate that the beneficial effect of Si in rising N content depends on $NH_4^+$ concentration in the medium, as reported in recent studies [6, 24, 25].

The addition of Si to the nutrient solution proved to be efficient in supplying the element to the plants, increasing its accumulation (Fig 1C), even though eucalyptus is not classified as an Si-accumulator [26].

In plant metabolism, N deficiency and $NH_4^+$-toxicity reduced the efficiency of photosystem II and increased energy loses throughout fluorescence (Fig 3). N deficiency is related to a decline in the $F_v/F_m$ ratio due to an increase in $F_0$ and $F_m$ in a number of species [27–29]. On the other hand, $NH_4^+$ toxicity alters the biochemical and physiological traits linked to changes in intracellular pH, phytohormone and polyamine metabolism, and greater oxidative stress, among other physicochemical modification [6].

Si can directly and indirectly favor photosynthetic reaction centers [30]. The element is related to greater cell wall stiffness, forming a double layer of Si in the leaf epidermis, thereby

improving leaf architecture and light absorption capacity, and resulting in less energy loss through fluorescence [31]. Silicon acts indirectly by decreasing physicochemical alterations, such as stomatal conductance and transpiration, which favor the photosynthetic rate [30], increasing the $F_v/F_m$ ratio under N deficiency and $NH_4^+$-toxicity.

SA foliar spray minimized the effects of high $NH_4^+$ concentration, reducing $F_0$ and maintaining a high $F_v/F_m$ ratio (Fig 3F). SA seems to be essential in sending signals that lead to an acquired systemic response, acting in the synthesis involved in producing defense compounds and regulating oxidative burst [32]. SA can protect specific proteins, decreasing post-stress lipid peroxidation, reflecting in greater quantum efficiency of photosystem II and higher electron transport efficiency ($\Phi PSII$), reducing energy loss by fluorescence [33].

The role of Si as elicitors was confirmed by the greater dry matter production of eucalyptus seedlings under N deficiency and $N-NH_4^+$ toxicity (Fig 3A and 3C). Si is absorbed by plants in the soluble form as monosilicic acid ($H_2SiO_4$) and not eliminated by the transpiration process, mitigating the damage caused to the leaf and root cell structure [34] and favoring the photosynthetic efficiency of plants, with a consequent increase in dry mass production, as demonstrated in eucalyptus (Figs 2 and 3) and other species [17, 24, 25, 35, 36].

The beneficial effect of SA, in turn, has been related primarily to the use of low hormone concentrations [37], where higher concentrations inhibit plant growth by lowering the photosynthetic rate and Rubisco activity [38, 39] The different responses are reported in terms of SA use in plants, and variations depend on the time of application, mode of use and environmental conditions. [40].

## 5. Conclusion

The hypothesis suggesting that the use of Si and SA mitigates N deficiency and ammoniacal toxicity in eucalyptus plants was accepted. As such, this information may prompt new strategies using Si via fertigation or SA foliar spraying in order to improve the sustainable production of eucalyptus seedlings that exhibit nitrogen-related nutrient disorder. Silicon supplied by fertigation and SA via foliar spraying mitigated damage caused by $NH_4^+$ deficiency and toxicity, favoring photosynthetic quality and increasing dry matter production.

## Author Contributions

**Conceptualization:** Jonas Pereira de Souza Junior, Renato de Mello Prado, Marcilene Machado dos Santos Sarah, Rinaldo César de Paula.

**Formal analysis:** Jonas Pereira de Souza Junior, Thaís chagas Barros de Morais, Marcilene Machado dos Santos Sarah.

**Investigation:** Jonas Pereira de Souza Junior, Thaís chagas Barros de Morais, Joaquim José Frazão, Marcilene Machado dos Santos Sarah, Kevein Ruas de Oliveira.

**Methodology:** Jonas Pereira de Souza Junior, Thaís chagas Barros de Morais, Joaquim José Frazão, Marcilene Machado dos Santos Sarah, Kevein Ruas de Oliveira.

**Project administration:** Jonas Pereira de Souza Junior, Renato de Mello Prado.

**Resources:** Renato de Mello Prado, Rinaldo César de Paula.

**Supervision:** Renato de Mello Prado.

**Writing – original draft:** Jonas Pereira de Souza Junior, Joaquim José Frazão, Kevein Ruas de Oliveira.

**Writing – review & editing:** Jonas Pereira de Souza Junior, Renato de Mello Prado, Rinaldo César de Paula.

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
