## [Decision Letter · Decision Letter 0]

24 Feb 2021

PONE-D-21-00654

Silicon fertigation and salicylic acid foliar spraying mitigate ammonium deficiency and toxicity in Eucalyptus spp. Clonal seedlings.

PLOS ONE

Dear Dr. Jonas,

Thank you for submitting your manuscript to PLOS ONE. After careful consideration, we feel that it has merit but does not fully meet PLOS ONE’s publication criteria as it currently stands. Therefore, we invite you to submit a revised version of the manuscript that addresses the points raised during the review process.

I agree with the reviewers that English editing should be done by native speaker. Material and methods section should be critically revised. Authors should provide a table of analysis of variance containing the mean squares and significance for each source of evaluation in each experiment.

Authors can elaborate more results especially about consequences of SA supplementation alone and together with Si.

We look forward to receiving your revised manuscript.

Kind regards,

Basharat Ali, Ph.D

Academic Editor

PLOS ONE

Journal Requirements:

"The funders had no role in study design, data collection and analysis, decision to publish, or preparation of the manuscript. "

3. Please upload a copy of Figure 4, to which you refer in your text on page 13 and 14. If the figure is no longer to be included as part of the submission please remove all reference to it within the text.

Reviewers' comments:

Reviewer's Responses to Questions

**Comments to the Author**

1. Is the manuscript technically sound, and do the data support the conclusions?

Reviewer #1: Yes

Reviewer #2: Yes

2. Has the statistical analysis been performed appropriately and rigorously? 

Reviewer #1: Yes

Reviewer #2: No

3. Have the authors made all data underlying the findings in their manuscript fully available?

Reviewer #1: Yes

Reviewer #2: Yes

4. Is the manuscript presented in an intelligible fashion and written in standard English?

Reviewer #1: Yes

Reviewer #2: No

5. Review Comments to the Author

Reviewer #1: The present study entitles “Silicon fertigation and salicylic acid foliar spraying mitigate ammonium deficiency and toxicity in Eucalyptus spp. Clonal seedlings” shows some interesting findings. However, there are few following concerns authors should answer before acceptance in your journal.

Abstract: Line 32 author need to remove one dot (.) and also check punctuation and other grammatical errors throughout the MS. Instead of using abbreviation (N) at the start of line write complete name Nitrogen content…Author can add more results in this section especially about consequences of SA supplementation alone and together with Si.

Introduction: This part is well written but in cases of Si and SA supplementation to improve the physiological attributes only cited one example for each. For the benefit of readers can cite more references i.e., Environ Sci Pollut Res (2016) 23:20483–20496, DOI 10.1007/s11356-016-7167-2; Ecotoxicology and Environmental Safety 133 (2016) 146–156;

Results and Discussion: In the results, line 157, please merge these two paragraphs into one (157-161 and 168-171) as it all about N. Similarly, in the next section (photosystem II) merge line 174-194 into one paragraph and merge 195-202 into one paragraph.

Reviewer #2: In my view, the manuscript has no innovation to be published in Plos One. If the Editor decides to request corrections, I send my suggestions below:

- How many leaves per treatment were used to evaluate (F0) and maximum (Fm) fluorescence and quantum efficiency of photosystem II (Fm / Fv)?

- Data analysis is very poor. The authors did not explore the correlation between the variables evaluated. Several multivariate techniques could be used to highlight these relationships and this has not been done.

- Authors need to provide a table of analysis of variance containing the mean squares and significance for each source of evaluation in each experiment. Include the CV values for each variable in this table.

- English in many parts is poor and needs a thorough review.

6. PLOS authors have the option to publish the peer review history of their article (what does this mean?). If published, this will include your full peer review and any attached files.

Reviewer #1: **Yes: **Rafaqat Ali Gill

Reviewer #2: No

---

## [Author Response · Author response to Decision Letter 0]

24 Mar 2021

Dear Prof. Basharat Ali

Academic Editor of PLOS ONE

We would like to thank the reviewers and the editor to the evaluation of our manuscript entitled “Silicon fertigation and salicylic acid foliar spraying mitigate ammonium deficiency and toxicity in Eucalyptus spp. clonal seedlings”. 

There was an improvement in the scientific quality of the reformulated text with the inclusion of corrections and suggestions of reviewers. All suggestions/corrections were carefully incorporated into the revised version of the manuscript, using the track changes mode in MS Word. Furthermore, we will be glad to introduce further changes if the editor believe necessary. Below there is a list of the modification introduced and our comments. 

Academic Editor – Observations

Thank you for submitting your manuscript to PLOS ONE. After careful consideration, we feel that it has merit but does not fully meet PLOS ONE’s publication criteria as it currently stands. Therefore, we invite you to submit a revised version of the manuscript that addresses the points raised during the review process. 

I agree with the reviewers that English editing should be done by native speaker. Material and methods section should be critically revised. Authors should provide a table of analysis of variance containing the mean squares and significance for each source of evaluation in each experiment. 

Authors can elaborate more results especially about consequences of SA supplementation alone and together with Si. 

Answer: We thank the editor for considering the merit of this research and indicating the need for a review.

Following the guidelines of the editor, we emphasize that:

An English native specialist, improving scientific writing, revised the entire text in relation to writing. The English revision certificate was sent with the material to PLOS ONE.

The review of the material and methods section was carefully performed following the points indicated by the reviewer 2. 

An analysis of variance table was added, with the squares of the means and the meanings for each evaluation and each experiment, as requested by the editor and reviewer 2.

It was evidenced in the manuscript that all the changes indicated by the editor were accepted by the authors and rigorously incorporated in the manuscript because we believe that the article had its scientific quality improved, being now in accordance with the criteria of PLOS ONE. 

Journal Requirements:

1. Please ensure that your manuscript meets PLOS ONE’s style requirements, including those for file naming. The PLOS ONE style templates can be found at https://journals.plos.org/plosone/s/file?id=wjVg/PLOSOne_formatting_sample_main_body.pdf and 

Answer: Following the guidance of the reviewer, indicated by the PLOS ONE’s website, we modified the title page and the text of the article to suit the PLOS ONE format style. We carefully review all the text in the requested format. 

a. Please clarify the sources of funding (financial or material support) for your study. List the grants or organizations that supported your study, including funding received from your institution.

d. If you did not receive any funding for this study, please state: “The authors received no specific funding for this work.”

Answer: We appreciate the clarification by the editor and declare that we do not receive specific funds for this word (Statement d). 

3. Please upload a copy of Figure 4, to which you refer in your text on page 13 and 14. If the figure is no longer to be included as part of the submission please remove all reference to it within the text.

Answer: The entire text was revised and “figure 4” was replaced by “figure 3”. 

Comments to the Author

Reviewer #1: The present study entitles “Silicon fertigation and salicylic acid foliar spraying mitigate ammonium deficiency and toxicity in Eucalyptus spp. Clonal seedlings” shows some interesting findings. However, there are few following concerns authors should answer before acceptance in your journal.

Answer: We thank the reviewer 1 for the comments indicating that the article reports interesting information and concerns before accepting the manuscript for publication. The reviewer’s guidelines were important to increase the scientific quality of the article, with no restriction for its publication in PLOS ONE

Abstract: Line 32 author need to remove one dot (.) and also check punctuation and other grammatical errors throughout the MS. Instead of using abbreviation (N) at the start of line write complete name Nitrogen content. Author can add more results in this section especially about consequences of SA supplementation alone and together with Si.

Answer: We thank reviewer 1 for the comments in the abstract session. 

A native English-speaking specialist, to correct punctuation and grammatical errors, revised the entire text. 

We replaced the abbreviation “N” with “Nitrogen” as indicated by the reviewer. 

To increase the scientific quality of the abstract, we added more results, as indicated by the reviewer, being the following sentence added to the text: “The addition of Si to the nutrient solution proved to be beneficial to the photosynthetic apparatus by decreasing F0 at 15 and 30 mmol L-1 of NH4+; and Fm at all NH4+ concentrations studied. In addition, the beneficial element also increases Fv/Fm at all NH4+ concentrations studied. SA foliar application was also efficient in reducing photosynthetic energy losses by decreasing F0 and Fm at all NH4+ concentrations studied. However, SA only increased the Fv/Fm at the high concentrations studied (30 and 60 mmol L-1 of NH4+). Nitrogen disorder by deficiency or N-NH4+ toxicity reduced shoot dry mass production. The addition of Si to the nutrient solution and SA foliar application increased shoot dry mass production at all N-NH4+ concentrations studied, and benefitted the photosynthetic apparatus by decreasing fluorescence and improving the quantum efficiency of photosystem II as well as dry mass production.” (L - 38-48).

Introduction: This part is well written but in cases of Si and SA supplementation to improve the physiological attributes only cited one example for each. For the benefit of readers can cite more references i.e., Environ Sci Pollut Res (2016) 23:20483–20496, DOI 10.1007/s11356-016-7167-2; Ecotoxicology and Environmental Safety 133 (2016) 146–156.

Answer: The reviewer’s suggestions was relevant. We included new and important references in the manuscript. 

Results and Discussion: In the results, line 157, please merge these two paragraphs into one (157-161 and 168-171) as it all about N. Similarly, in the next section (photosystem II) merge line 174-194 into one paragraph and merge 195-202 into one paragraph.

Answer: As requested by the reviewer, we merge the requested paragraphs. 

Awnser: Atendendo ao solicitado pelo revisor, juntamos os parágrafos solicitados. 

Reviewer #2: In my view, the manuscript has no innovation to be published in Plos One. If the Editor decides to request corrections, I send my suggestions below:

Answer: We understand the criticism of the reviewer 2 indicating lack of innovation in the research, as there is information in the literature with emphasis on silicon and ammonium. However, it is worth considering that there are no reports on eucalyptus seedlings (Eucalyptus urophylla x Eucalyptus grandis) indicating the plant’s physiological response involving ammonium and silicon/salicylic acid. It is also verified that the research results are well scientifically substantiated. Our research should fill this gap with benefit on a global scale because the production of eucalyptus seedlings on an inert substrate that uses ammoniacal nitrogen occurs in many countries that grow this species. We take the opportunity to thank the reviewer 2 for the suggestions, as we know that he aims to contribute to the quality of the manuscript and we will comment on them below. 

- How many leaves per treatment were used to evaluate (F0) and maximum (Fm) fluorescence and quantum efficiency of photosystem II (Fm / Fv)?

Answer: Photosynthetic evaluations were performed on the second pairs of leaves completely developed in each plant (as indicated in the text – L 36).

To increase clarity for readers, we have included the following sentences in the text: “The experiments were concluded 30 days after seedling transplanting, at which time visual signs of N toxicity, in its NH4+ forms, were identified, such as chlorosis, necrosis, brown stem and wilted leaves displaying signs of senescence [18]. On this occasion, in both experiments, the initial (F0) and maximum (Fm) fluorescence and quantum efficiency of photosystem II (Fv/Fm) on the second pair of fully expanded leaves were measured. These variables were obtained by measuring chlorophyll fluorescence in twelve leaves (6 repetitions and two plants per repetition) in experiment I; and eighteen leaves (6 repetitions and three plants per repetition) in experiment II, using a fluorometer (Opti-Science – OS30P). These measurements were taken at the end of the experiment, between 7:30 and 9:30 am, and on three new fully formed leaves (middle part of the stalk) per plant. Leaves were left in the dark for 30 min for adaptation purposes and then excited by a pulse of red light for 1 second, in order to determine the initial, maximum and the variable fluorescence. The quantum efficiency of photosystem II was calculated by the ratio between maximum and variable fluorescence.” (L - 132-144) 

- Data analysis is very poor. The authors did not explore the correlation between the variables evaluated. Several multivariate techniques could be used to highlight these relationships and this has not been done.

Answer: We respect the reviewers comment. We take the opportunity to ponder. The statistical analysis used in the manuscript is well described in the material and methods of the manuscript (L – 160-163). It was defining polynomial regression analysis for quantitative studies (increasing ammonium concentration) and comparison test of means for studying the absence and presence of silicon/salicylic acid. These analyzes were sufficient to verify the effects of the treatments studied on the assessed physiological aspects. We believe that it is unnecessary to include correlation analysis between the variables as the results would be predictable and would contribute little to the scientific quality of the manuscript. For example, the effect of a treatment on the increase in Fv/Fm should increase the production of dry mass, therefore, a high correlation of these two variable is expected, thus, it would have limited contribution to a better understanding of the results obtained. However, if the editor deems it necessary to add a new statistical analysis to the results obtained from the experiments, we are available. 

- Authors need to provide a table of analysis of variance containing the mean squares and significance for each source of evaluation in each experiment. Include the CV values for each variable in this table.

Answer: Following the guidance of the reviewer 2 and the editor, we added the analysis of variance table to the text with the values of the indicated coefficient of variation. 

- English in many parts is poor and needs a thorough review.

Answer: Following the guidance of the reviewers ant the editor, the entire text was revised by native English speaker. The revision certificate is attached.

---

## [Decision Letter · Decision Letter 1]

7 Apr 2021

Silicon fertigation and salicylic acid foliar spraying mitigate ammonium deficiency and toxicity in Eucalyptus spp. Clonal seedlings.

PONE-D-21-00654R1

Dear Dr. Jonas Pereira Souza Junior,

We’re pleased to inform you that your manuscript has been judged scientifically suitable for publication and will be formally accepted for publication once it meets all outstanding technical requirements.

Kind regards,

Basharat Ali, Ph.D

Academic Editor

PLOS ONE

Additional Editor Comments (optional):

Reviewers' comments:

Reviewer's Responses to Questions

**Comments to the Author**

1. If the authors have adequately addressed your comments raised in a previous round of review and you feel that this manuscript is now acceptable for publication, you may indicate that here to bypass the “Comments to the Author” section, enter your conflict of interest statement in the “Confidential to Editor” section, and submit your "Accept" recommendation.

Reviewer #2: (No Response)

2. Is the manuscript technically sound, and do the data support the conclusions?

Reviewer #2: (No Response)

3. Has the statistical analysis been performed appropriately and rigorously? 

Reviewer #2: (No Response)

4. Have the authors made all data underlying the findings in their manuscript fully available?

Reviewer #2: (No Response)

5. Is the manuscript presented in an intelligible fashion and written in standard English?

Reviewer #2: (No Response)

6. Review Comments to the Author

Reviewer #2: (No Response)

7. PLOS authors have the option to publish the peer review history of their article (what does this mean?). If published, this will include your full peer review and any attached files.

Reviewer #2: No

---

## [Editor Report · Acceptance letter]

12 Apr 2021

PONE-D-21-00654R1 

Silicon fertigation and salicylic acid foliar spraying mitigate ammonium deficiency and toxicity in *Eucalyptus* spp. clonal seedlings. 

Dear Dr. de Souza Junior:

I'm pleased to inform you that your manuscript has been deemed suitable for publication in PLOS ONE. Congratulations! Your manuscript is now with our production department. 

Kind regards, 

on behalf of

Dr. Basharat Ali 

Academic Editor

PLOS ONE